# Synthesis and Performance Analysis of Green Water and Oil-Repellent Finishing Agent with Di-Short Fluorocarbon Chain

**DOI:** 10.3390/molecules28083369

**Published:** 2023-04-11

**Authors:** Yanli Li, Yi Luo, Qinqin Wang, Wei Zou, Wenjiang Zheng, Xiaoyan Ma, Hu Yang

**Affiliations:** School of Chemical Engineering, Sichuan University of Science and Engineering, Zigong 643000, China

**Keywords:** water-repellent finishing agent, short fluorocarbon chain, emulsion polymerization, contact angle

## Abstract

A novel fluorine-containing water-repellent agent (OFAE-SA-BA) was designed and synthesized by emulsion copolymerization, which was used to replace the commercial long fluorocarbon chain water-repellent agent. To improve water repellency, the intermediate and monomer containing two short fluoroalkyl chains were successfully synthesized and characterized by ^1^H NMR, ^13^C NMR and FT-IR, respectively. After being treated by the water-repellent agent, the surface chemical composition, molecular weight, thermal stability, surface morphology, wetting behavior, and durability of the modified cotton fabrics were characterized using X-ray photoelectron spectrophotometry (XPS), gel permeation chromatography (GPC), thermal degradation (TG), scanning electron microscopy (SEM), and video-based contact angle goniometry, respectively. The cotton fabric demonstrated water contact angle of 154.1°, both the water and oil repellency rating were grade 4. The durability of water repellency of the treated fabrics only decreased slightly after 30 times, which represented very good washing durability. The finishing agent did not affect the whiteness of the fabric.

## 1. Introduction

Due to the great potential for application in scientific and industrial areas, the fabrication of superhydrophobic surfaces on textile substrates, which possesses unique waterproofing and self-cleaning properties, has attracted great interest recently for both academic research and industrial applications [1,2,3]. Water-repellent finishing is an important process in the textile industry. Hydrophobic and oleophobic fabrics are in great demand in a variety of functional applications including curtains, outdoor gear, rainwear, stain- resistant products, vehicle interiors, bandages, etc. [4,5,6,7,8]. A fabric finished with fluoropolymers not only has excellent water, oil, and stain repellency but is also able to maintain its original color, texture, and wearing comfort. Therefore, the fluorine-containing acrylate polymer has obtained a wide range of applications in textile finishing.

In the past decades, it has been well known that organic perfluorinated compounds with a long perfluoroalkyl chains (R_fn_, n > 8) have excellent hydrophobic and oleophobic effects, especially as perfluorooctane sulphonate (PFOS) and perfluorooctanoic acid (PFOA) have already been widely used in the field of fabric dyeing and finishing [9,10]. However, there is increasing evidence that indicates that compounds containing long perfluoroalkyl chains are harmful to human beings and animals owing to their high bioaccumulation, difficult biodegradation, and long-distance migration [11,12,13]. Therefore, their application has already been restricted by the European Union commission [14]. The USA and Canada have also instituted several bans that aim to eliminate or restrict their production and use [15,16]. In the face of the worldwide prohibition of PFOS and PFOA, the development of environment-friendly alternative fluorinated materials to substitute for traditional reagents associated with PFOS/PFOA is urgent [17,18,19,20].

More researchers have introduced the short perfluoroalkyl chain (-C_n_F_2n+1_, n ≤ 6) into the fabric finishing agent to achieve the water- and oil-repellent finishing of textiles [21,22,23]. The short perfluoroalkyl chain compounds do not have the obvious properties of persistence and bioaccumulation; moreover, they will be metabolized and excreted in a short time, and their degradation products are non-toxic [24,25,26]. The widely used product Scotchgard Protector was manufactured with perfluorobutyl sulphonate (PFBS) produced by 3M, which was used to replace PFOS. Additionally, the water- and stain-repellency of PFBS is good, but its oil-repellency is poor; the principal reason is its short perfluoroalkyl chain (n = 4) [27]. Perfluorinated acrylate polymer has extremely low surface energy, which leads to its application in high-performance coatings and textiles, as a leather finishing agent, and in other areas [6]. Novel multifunctional graft copolymers containing short fluoroalkyl side chains and reactive groups that were successfully prepared by grafting fluoroalcohols to polyacrylate exhibited relatively low water repellency and had no oil repellency [28]. A new kind of multifunctional polyacrylate latex containing fluorinated and hydrophilic groups (FHPA) was synthesized by semi-continuous emulsion polymerization, and the water and oil repellent had been greatly improved [29]. However, textile finishing agents with short fluorocarbon chains usually exhibited hydrophobic and oleophobic properties, which would lead to a dramatic decline, making them impractical [30].

To solve this problem, multiple short fluorocarbon chains were introduced into polyacrylates to improve the hydrophobicity and oleophobicity and avoid the environmental problems. Octafluoropentanol was used as a short fluorocarbon chain to synthesize a fluorine-containing functional monomer; butyl acrylate and octadecyl acrylate were used as soft monomers; and a finishing agent could be achieved via copolymerization, which was then coated onto the fabric. The surface and application properties of the coated fabric were then characterized and evaluated.

## 2. Results and Discussion

### 2.1. Synthesis of Finishing Agent Fluorine-Containing Polymer with Short Chain

#### 2.1.1. Synthesis of Intermediate 2,3-Bisoctafluoropentyloxy-1-propanol (OFPA)

The synthesis route of OFPA is shown in Figure 1. The chemical structure of the resultant OFPA was characterized by ^1^H NMR (Figure 2), ^13^C NMR (Figure 3) and ^19^F NMR (Figure 4), respectively. As shown in Figure 2, the typical signal of -OH was found at 4.01 ppm. The peak observed at 3.70 ppm was assigned as the hydrogen atoms of -CH_2_-O-CH_2_-. The peak with chemical shift at 2.33 ppm arose from -CH-. The peak at 6.05 ppm originated from -CF_2_H. In the ^13^C NMR spectrum of OFPA (Figure 3), the resonance at 105–118 ppm corresponded to the fluorocarbon chain. The peaks at 70 ppm and 73 ppm originated from the carbon, which linked the ether bond, and the peak at 68 ppm was considered the carbon-linked hydroxyl. As shown in Figure 4 of ^13^F NMR spectrum, the peaks observed at -120.00 ppm, -125.73 ppm, -130.39 ppm and -137.38 ppm were assigned as the fluorine atoms of HF_2_CF_2_CF_2_CF_2_C-, respectively. MS (ESI) m/z: 519.0 (A refers to the molecular weight of OFPA. Calculated for {[A]-} 520.21). ^1^H NMR, ^13^C NMR, ^19^F NMR and mass spectrometry analysis mutually confirmed that intermediate OFPA was successfully synthesized.

#### 2.1.2. Synthesis of Monomer 2,3-Bisoctafluoropentyloxy-1-acrylate (OFAE)

The synthesis route of OFAE is shown in Figure 5. In order to characterize the structure of the esterified OFPA, OFAE was also analyzed by ^1^H NMR, ^13^C NMR and ^19^F NMR. The ^1^H NMR spectrum was presented in Figure 6. In the ^1^H NMR spectra of OFAE, the spectra showed new signals at 6.44 ppm, 5.89 ppm, and 6.12 ppm that could be attributed to -CH=CH_2_. The peak with chemical shift at 6.02 ppm was -CF_2_H. The peaks at 3.98 ppm and 3.94 ppm corresponded to -CH_2_-O-CH_2_-. Resonance signals at 5.19 ppm were assigned to -CH-. Figure 7 provides the ^13^C NMR spectrum of OFAE. In the ^13^C NMR spectrum of OFAE, the resonances in the region of 105–120 ppm were attributed to the fluorocarbon chain, the spectrum revealed the ester bond at 165 ppm, and the peaks at 127 ppm and 132 ppm corresponded to the olefin. The peaks at 70 ppm and 68 ppm originated from the carbons of the ether bond. The ^19^F NMR spectrum was displayed in Figure 8, the peaks observed at -120.02 ppm, -125.73 ppm, -130.42 ppm and -137.39 ppm were as-signed as the fluorine atoms of HF_2_CF_2_CF_2_CF_2_C-, respectively. MS (ESI) m/z: 574.05 (A refers to the molecular weight of OFAE calculated for {[A]-} 574.06). ^1^H NMR, ^13^C NMR, and mass spectrometry analysis mutually confirmed that monomeric OFAE was successfully synthesized.

#### 2.1.3. Fourier Transform Infrared Spectra Analysis of Water-Repellent Fabric Finishing Agent (OFAE-SA-BA)

Fourier transform infrared spectrometry was used to characterize the chemical structure of the fabric finishing agent OFAE-SA-BA, and the corresponding FT-IR spectrum was presented in Figure 9. The peaks at 2962–2877 cm^−1^ and 1458 cm^−1^ were attributed to symmetric stretch and asymmetric stretch of -CH_2_-. The strong peak that appeared at 1734 cm^−1^ was attributed to the stretching vibration of carbonyl C=O group. The band at 1250 cm^−1^ represented the characteristic peaks of ether C-O-C. The peaks at 1171 cm^−1^ and 1129 cm^−1^ represented the symmetric vibration mode of -CF_2_-. In conclusion, the results indicate that the target product was indeed the fabric finishing agent.

### 2.2. XPS Analysis

The surface elemental composition of the fabrics was ascertained by XPS. Figure 10 shows the survey scans of cotton fabric surfaces before and after OFAE-SA-BA finishing. As shown in Figure 10A, the pristine commercial cotton fabric only possessed C1s and O1s at 284 eV, 532 eV, respectively. The atomic content of C1s on its surface could reach as high as 73.47% (Table 1). After treatment with an oil-repellent fabric finishing agent (Figure 10B), the atomic content of F1s on the surface was increased up to 25.66%.

### 2.3. Gel Permeation Chromatography (GPC) Analysis

GPC is one of the most widely used techniques for the determination of molecular weight (Mw), number average molecular weight (Mn), and polydispersity index (PDI) of OFAE-SA-BA. PDI and Mw are listed in Table 2. Mw and Mn of OFAE-SA-BA were approximately 3.45 × 10^5^ and 3.64 × 10^5^, respectively. Additionally, the PDI was 1.05. These indicated that the Mw was in the normal ranges and there was little crosslinking, according to the PDI.

### 2.4. Thermal Degradation Property

The thermal stability of OFAE-SA-BA was evaluated through the TGA experiments from 25 to 800 °C at a heating rate of 10 °C/min. The TG curve is shown in Figure 11. It is clear that the pyrolysis of the polymer included only one mass loss step. The thermal degradation of the polymer started in the range of 230–300 °C. OFAE-SA-BA exhibited a temperature of 400 °C at 50% weight loss under nitrogen. The final weight loss of polymer was nearly 97% at approximately 525 °C. This indicates that the weight loss corresponded to the main polymer chain decomposition; OFAE-SA-BA was thermally stable within the temperature range for its applications.

### 2.5. Particle Size and Stability of Fabric Finishing Agent

The average particle size of the fabric finishing agent was 120.5 nm, which indicated that the finishing agent emulsion had a relatively narrow particle distribution and small particle size; therefore, polymer particles may have little tendency for aggregation and sedimentation and thus would have good stability. The storage stability of the finishing agent was confirmed by storing it at room temperature for 1 month. The polymer emulsion samples had not been stratified or precipitated by the stability test, which indicated that the storage stability was satisfactory.

### 2.6. Surface Morphology of Cotton Fibers

The surface morphology of cotton fibers before and after modification was characterized by SEM in Figure 12. As can be seen in Figure 12a,b, it could be clearly found that the surface of the unfinished cotton fibers was relatively rough; there were some uneven and slender grooves on the surface. On the contrary, as shown in Figure 12c,d, after the fabric was coated by the polymer, an evidently even film was markedly generated on the surface of the fiber, and the deep grooves of the treated fabric were virtually concealed by the film so that the surface became smoother compared to the pristine fiber.

### 2.7. Evaluation of Hydrophobicity Property

The surface wettability, generally governed by the combination of surface chemical composition and suitable microstructure, was commonly examined by the water contact angle measurement. Accordingly, water- and oil-repellency rating measurements were used to investigate the wettability of the treated cotton fabrics as a supplement. A higher rating indicates less wettability [31].

Figure 13a,c presents the resulting spraying properties of untreated and treated fibers, and commensurable static contact angles are exhibited in Figure 13b,d. It can be seen that when a water droplet was dripped onto the untreated cotton fabric, it could spread rapidly on the surface and the fabric would be completely wet (Figure 13a). Additionally, the water contact angle (WCA) of the unfinished cotton fabric was only 71.7° (Figure 13b); there was no waterproof grade at all, according to AATCC22-2018. When the pristine fabric was coated by the agent, these values were dramatically improved and revealed a good repellency towards water. As demonstrated in Figure 13c, the water drops rolled off the surface, which was slightly wet, the water-repellency rating greatly improved to the fourth grade; meanwhile, water droplets could stay on the finished cotton fibers, WCA was significantly increased to 154.1°, as shown in Figure 13d. The average water contact angle of finished cotton fibers was above 150°. The surface contact angle was greater than 150°, indicating that the substrates favored superhydrophobicity [32].

When the treated fabric was in the air, close-packed fluorocarbon chains dominated the surface and presented a very low surface energy, while the hydrophilic groups were collapsed below the surface; therefore, water tend to be repelled. Compared with the existing marketable water repellents with C6 short-fluorinated finishing agents, the WCA of the cotton fabrics which were finished with three marketable finishing agents, according to the same finishing and testing conditions, were 143.8°, 145.2°, and 144.7°, respectively [33]. The water repellency of OFAE-SA-BA was superior to that of the three marketable finishing agents; meanwhile, the fluorine chains were shorter, with only C4, which was more environmentally friendly than C_6_ [34].

The oil-repellency rating of the treated cotton fabric is shown in Figure 14. According to AATCC118-2013, oleophobicity has been investigated by evaluating the oil-repellent rating in the range of 0~8, according to the final morphological characteristics of eight typed standard oil droplet fluids on the surface of cotton fabric. In the comparison of three samples, the untreated cotton fabric (Figure 14a) was thoroughly wetted by level 1 standard fluid white mineral oil, which demonstrated that it had no oil repellency. As demonstrated in Figure 14c, level 3 standard fluid n-hexadecane exhibited an almost spherical shape and no surface wetting was observed. Moreover, for level 4 standard fluid n-tetradecane there was no apparent wetting and the morphology of the oil droplet remained ellipsoidal. The rating of oil repellency could be determined as 4. Although the oil repellent effect was still not ideal, other C6 short-fluorinated finishing agents showed almost no oil repellency.

### 2.8. Durability of Water-Repellency

As an excellent finishing agent, in addition to adequate performance in functionality, durability is also important in practical applications. In order to evaluate washing durability, the treated cotton fabric samples were washed for different cycles (from 5 to 30 times), and the experiments were carried out according to the test method of AATCC 124-2014. The wash resistance of treated cotton fabrics (from 5 to 30 times) was evaluated, and the water contact angle and water- and oil-repellency ratings were also assessed, as shown in Table 3. After washing five times, the water contact angle decreased from 156.7° to 148.9°, and the water- and oil-repellency ratings had not changed, remaining at 4 and 4, respectively. This indicated that the treated cotton fabric can resist washing for several times, and was only minimally changed in terms of water- and oil-repellency performance. Even when it was washed 30 times, the water contact angle only decreased to 139.8°, and the water- and oil-repellency ratings dropped to 3 and 3, respectively. These results showed there was not large reduction in water and oil repellency compared with unwashed sample. Traditional fluorinated finishing agents do not always show sufficient stability during use because the coating materials are only attached to the top of the fabrics through physical adsorption or adhesion of the coatings to the textile substrate. It may be that OFAE-SA-BA was firmly bonded to the surface or interior of the cotton fibers by covalent crosslinking between the hydroxyl groups of the cotton fibers and the copolymer molecules, which led to good durability.

### 2.9. Whiteness

The whiteness of all fabrics was tested more than 5 times. The whiteness of untreated fabrics was 92.1 ± 1.24, and the whiteness of treated fabrics was 91.7 ± 1.45. There was no obvious difference in whiteness between untreated and treated fabrics.

## 3. Materials and Methods

### 3.1. Materials

Octafluoropentanol was supplied by the Zhong Hao Chen Guang Research Institute of Chemical Industry Co., Ltd. (Zigong, China). Acryloyl chloride, azo-bis-isobutyronitrile (AIBN), and deuterated chloroform (CDCl_3_) were supplied by Sigma Aldrich (Shanghai, China). Toluene, epichlorohydrin, xylene, sodium hydroxide, pyridine, sodium hydroxide (NaOH), butyl acrylate (BA), stearyl acrylate (SA), tetrahydrofuran (THF), methanol, ethyl acetate, terabutyl ammonium bromide, triethylamine, dichloromethane (CH_2_Cl_2_), sodium sulphate (NaSO_4_), alkylphenol ether sulfosuccinate sodium salt (MS-1), and sodium dodecyl sulfonate (SDS) were obtained from Sinopharm Co. (Shanghai, China).

### 3.2. Synthesis of Finishing Agent Fluorine-Containing Polymer with Short Chains

#### 3.2.1. Synthesis of Intermediate 2,3-Bisoctafluoropentyloxy-1-propanol (OFPA)

The synthesis route of OFPA is shown in Figure 1. In a 100 mL flask, NaOH (4.6 g, 115 mmol) was added to a solution of octafluoropentanol (24.37 g, 110 mmol) and terabutyl ammonium bromide (0.1 g, 0.312 mmol) in toluene (50 mL). After stirring for 30 min at room temperature, epichlorohydrin (4.63 g, 50 mmol) dissolved in 10 mL toluene was added dropwise. The resulting mixture was stirred at 73 °C for 12 h. When the starting materials disappeared upon inspection by TLC, the reaction mixture was washed with distilled water (50 mL × 3). The organic phase was dried with anhydrous sodium sulfate, followed by filtration and concentration under reduced pressure. After molecular distillation, a colorless and transparent liquid 2,3-Bisoctafluoropentyloxy-1-propanol (OFPA) (19.37 g) was obtained, with a yield of 74.5%. ^1^H NMR (600 MHz, CDCl_3_): δ (ppm) = 6.05 (2H, -CF_2_H), 4.01 (5H, -CH_2_-O-CH_2_-CHOH-CH_2_-O-CH_2_-), 3.70 (4H, -O-CH_2_-CHOH-CH_2_-O-), 2.33 (1H, -O-CH_2_-CH-CH_2_-O).^13^C NMR (600 MHz, CDCl_3_): δ (ppm)= 68.04~68.38 (-O-CH_2_-CHOH-CH_2_-O-), 69.29 (-O-CH_2_-CHOH-CH_2_-O-), 73.41 (-O-CH_2_-CF_2_-), 105.93~117.04 (-CF_2_-CF_2_-CF_2_-CF_2_H). ^19^F NMR (600 MHz, CDCl_3_, ppm): δ = -120.00 (HF_2_CF_2_CF_2_CF_2_C-), -125.73 (HF_2_CF_2_CF_2_CF_2_C-), -130.39 (HF_2_CF_2_CF_2_CF_2_C-), -137.38 (HF_2_CF_2_CF_2_CF_2_C-).

#### 3.2.2. Synthesis of Monomer 2,3-Bisoctafluoropentyloxy-1-acrylate (OFAE)

The synthesis route of OFAE is shown in Figure 5. To a stirred solution of OFPA (15.8 g, 30.4 mmol) in absolute dry THF (40 mL), triethylamine (3.54 g, 35 mmol) was added under N_2_ atmosphere. Acryloyl chloride (2.9 g, 32 mmol) dissolved in THF (10 mL) was slowly added dropwise into the reaction solution and stirred at room temperature for 10 h. After the reaction was completed, the solvent was removed in vacuo. The residue was poured into dichloromethane (50 mL) and washed with sodium bicarbonate solution (5 wt.%) to pH = 7~8. The combined organic layer was dried with anhydrous NaSO_4_, then filtered and evaporated in vacuo to yield monomeric OFAE as a yellow liquid (13.72 g, 78.66% yield). ^1^H NMR (600 MHz, CDCl_3_): δ (ppm) = 6.44 and 5.89 (2H, -CH=CH_2_), 6.16 (1H, -CH=CH), 6.12~5.93 (2H, -CF_2_H), 5.17 (1H, -O-CH_2_-CH(-O-C=O-C=CH_2_)-CH_2_-O), 4.03~3.91 (4H, -O-CH_2_-CF_2_-), 3.92 (4H, -CH(O)-CH_2_-O-CH_2_-CF_2_-). ^13^C NMR (600 MHz, CDCl_3_): δ (ppm)= 105~120 (-CF_2_-CF_2_-CF_2_-CF_2_H), 70.52 (-CF_2_-CH_2_-O-), 69.20 (-O-CH_2_-CH(O)-CH_2_-O-), 165 (-C=O-), 132 (-C=CH_2_-), 127 (-C=CH_2_-). ^19^F NMR (600 MHz, CDCl_3_, ppm): δ = -120.02 (HF_2_CF_2_CF_2_CF_2_C-), -125.73 (HF_2_CF_2_CF_2_CF_2_C-), -130.42 (HF_2_CF_2_CF_2_CF_2_C-), -137.39 (HF_2_CF_2_CF_2_CF_2_C-).

#### 3.2.3. Synthesis of Water- and Oil-Repellent Fabric Finishing Agent Octafluoropentyloxy Polyacrylate (OFAE-SA-BA)

First, a monomeric mixture of OFAE (5.6 g, 9.9 mmol), SA (3.18 g, 9.8 mmol), and BA (0.63 g, 4.9 mmol) were dissolved in 20 mL THF solvent. Then, the emulsifiers MS-1 (0.33 g) and SDS (0.33 g), melted in 20 mL deionized water, were protected by dry N_2_ and emulsified with 10,000 r/min cutting speed in an emulsifying machine. The mixture of monomers and the aqueous solution of AIBN (0.15 g) were added to the emulsion at 65 °C for 16 h. Finally, the reaction was stopped and cooled to obtain the target polymer, the so-called water- and oil-repellent fabric finishing agent octafluoropentyloxy polyacrylate (OFAE-SA-BA). Yield: 92%. ATR-IR: 2962, 2877, 1734, 1457, 1250, 1171, 1129, 1065, 964 cm^−1^.

### 3.3. Treatment of Cotton Fabrics

The cotton fabrics were immersed in the OFAE-SA-BA solutions (30 g/L, bath ratio 1:20) for 15 min and padded with two dips and nips with a wet pick-up of 90%. After the treatment, the cotton fabrics were dried at 80 °C for 3 min and cured at 160 °C for 3 min.

### 3.4. Characterization

#### 3.4.1. NMR Spectroscopy

Nuclear magnetic resonance (^1^H NMR and ^13^C NMR) spectra of the intermediate and monomer using deuterated chloroform (CDCl_3_) as a solvent and tetramethylsilane (TMS) as an internal standard were recorded with an AVANCE III HD 600 MHz spectrometer (Bruker Co., Ettlingen, Germany) under room temperature.

#### 3.4.2. Fourier Transform Infrared Spectra (FT-IR) Analysis

The IR spectra were recorded on a Nicolet-5700 Fourier transform infrared spectrometer (Thermo Fisher Scientific, Waltham, MA, USA) with a disc of KBr in the region of 4000 to 400 cm^−1^ and with a resolution of 1 cm^−1^.

#### 3.4.3. Mass Spectrum Analysis

The samples were prepared as CH_3_CN solutions. The molecular weights were measured by LC/MS 7890 A (Agilent, Santa Clara, CA, USA).

#### 3.4.4. Particle Size Analysis

The particle size of the finishing agent was measured by Litesizer 500 (Anton Paar, Graz, Austria).

#### 3.4.5. X-ray Photoelectron Spectroscopy (XPS)

The chemical composition of the cotton fabrics before and after finishing was analyzed by XPS (K-Alpha Thermo Electron Corporation, Santa Clara, CA, USA), with monochromatic Al K radiation (200 W, 12 KV, 1486.68 eV) employed to determine the surface chemical composition of the polyester fabrics.

#### 3.4.6. Gel Permeation Chromatography (GPC)

A Shimadzu LC20 gel permeation chromatographer (GPC) with a refractive index detector was used to measure the molecular weight (Mw). Linear polystyrene was used as the standard and THF as the solvent.

#### 3.4.7. Thermogravimetry Analysis (TGA)

The thermal stability of the polymer was studied using a TGA Q500 thermogravimetric analyzer (TA Instrument, New Castle, DE, USA). The heating rate was 10 °C/min and it was heated to 800 °C in a nitrogen atmosphere. The weight of the sample was approximately 10 mg during the heating process. The sample was dried at 110 °C to remove any moisture prior to the study.

#### 3.4.8. Scanning Electron Microscope (SEM) Analysis

Surface morphology characteristics of untreated and treated cotton fabric samples were performed on a SU1510-type field emission scanning electron microscope (SEM) (Hitachi, Japan).

#### 3.4.9. Hydrophobicity Test

The wettability of the cotton fabric samples was evaluated by contact angle measurement. Water contact angles (WCA) before and after treated cotton fabric surfaces were estimated using a DSA25 contact angle goniometer (Kruss, Hamburg, Germany). The contact angle value was acquired 10 s after the dropping of a distilled water drop (about 5 μL) onto the fabrics. The contact angle of each sample was measured at least three times and then averaged.

#### 3.4.10. Whiteness Test

The whiteness of untreated and treated cotton fabrics was measured according to standard GB/T 8424.2-2001 [35].

## 4. Conclusions

In summary, OFAE-SA-BA, which is a fluorine-containing water-repellent finishing agent of polyacrylate with multiple short fluorocarbon chains, was synthesized by means of emulsion copolymerization, with 2,3-bisoctafluoropentyloxy-1-acrylate (OFAE) as a fluorine-containing functional monomer. The intermediate and monomer were successfully synthesized and characterized by ^1^H NMR and ^13^C NMR; the polymer was characterized by FT-IR, and its molecular weight was 3.6 × 10^5^ by GPC. The agent was applied to the finishing of cotton fabrics. XPS analysis confirmed that OFAE-SA-BA had been successfully applied onto the surface of the cotton fabric. Additionally, OFAE-SA-BA presented good thermal stability and was thermally stable within the temperature range for its applications. The SEM photo showing the morphology before and after treatment with OFAE-SA-BA indicates that the agent could form a film on the surface of the cotton fibers, which resulted in the cotton fibric showing superhydrophobicity; the water- and oil- repellency ratings were both graded as a 4 on the grading scale. The water-repellency test after the washing revealed that OFAE-SA-BA demonstrated good persistence.

## Figures and Tables

**Figure 1 molecules-28-03369-f001:**
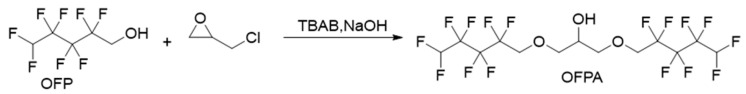
Synthesis route of OFPA.

**Figure 2 molecules-28-03369-f002:**
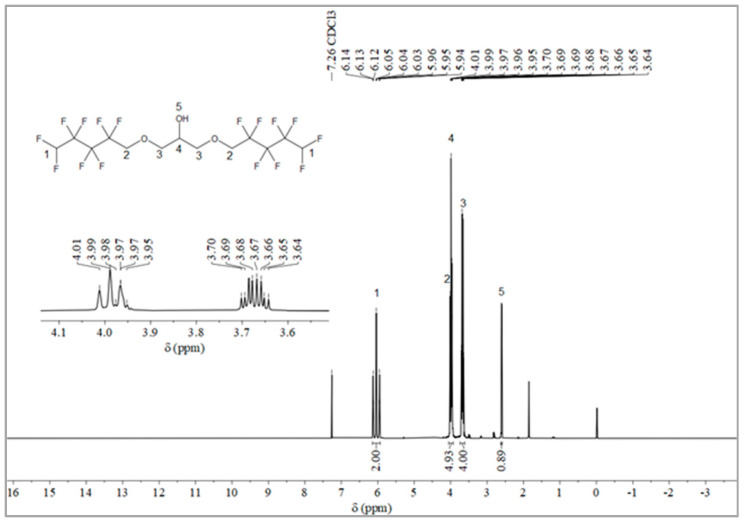
^1^H NMR spectra of OFPA.

**Figure 3 molecules-28-03369-f003:**
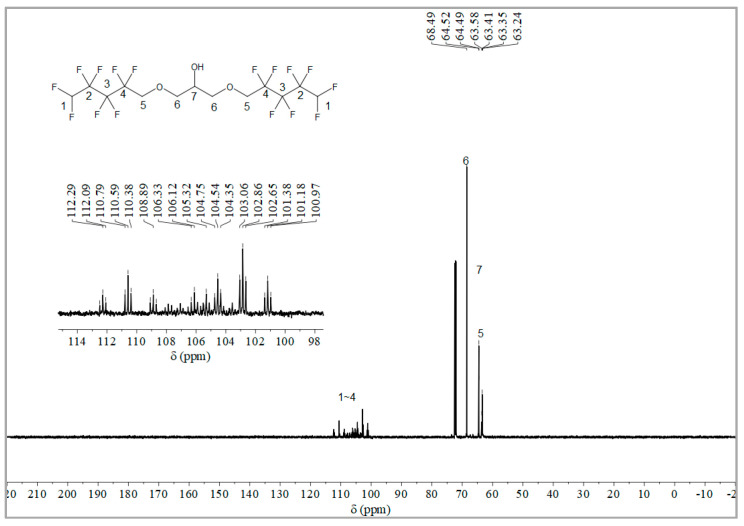
^13^C NMR spectra of OFPA.

**Figure 4 molecules-28-03369-f004:**
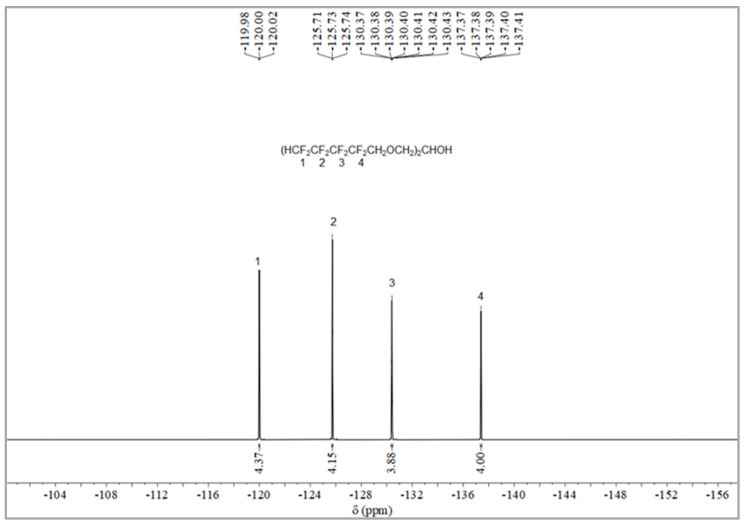
^19^F NMR spectra of OFPA.

**Figure 5 molecules-28-03369-f005:**
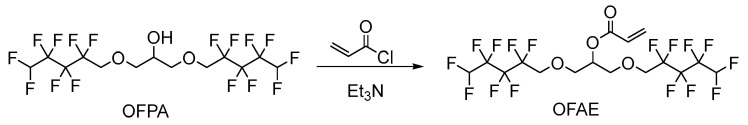
Synthesis route of monomeric OFAE.

**Figure 6 molecules-28-03369-f006:**
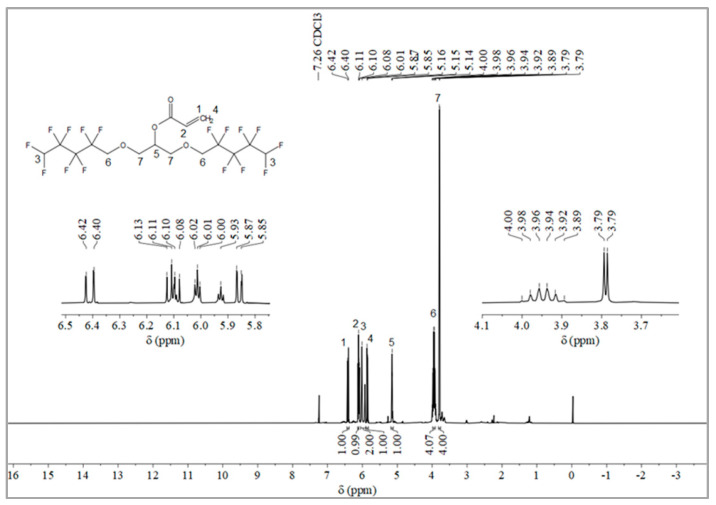
^1^H NMR spectra of OFAE.

**Figure 7 molecules-28-03369-f007:**
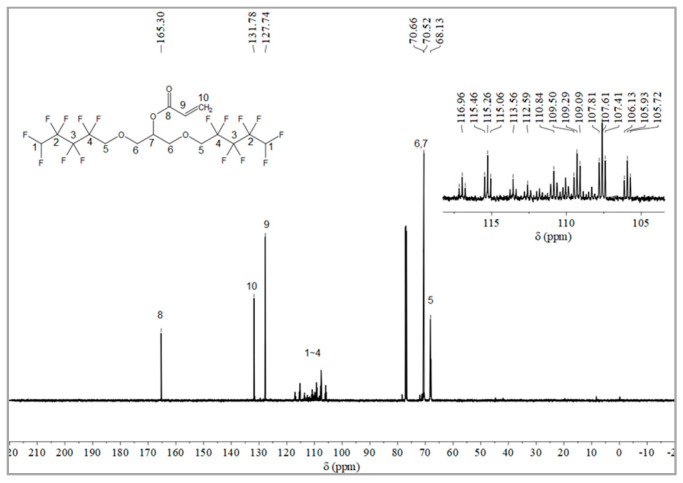
^13^C NMR spectra of OFAE.

**Figure 8 molecules-28-03369-f008:**
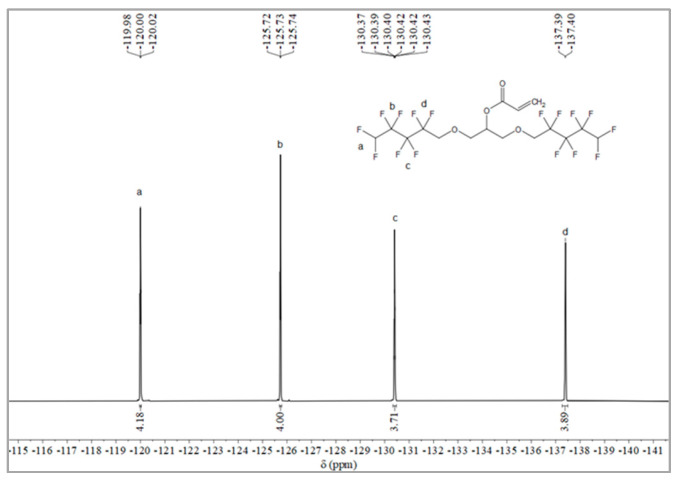
^19^F NMR spectra of OFAE.

**Figure 9 molecules-28-03369-f009:**
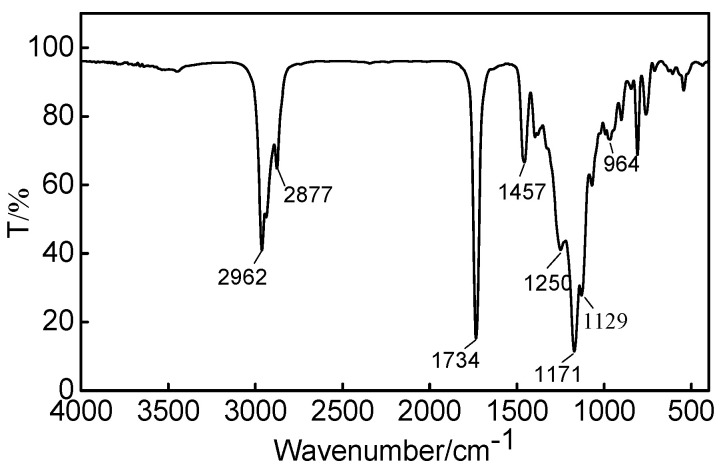
FT-IR spectrum of OFAE-SA-BA.

**Figure 10 molecules-28-03369-f010:**
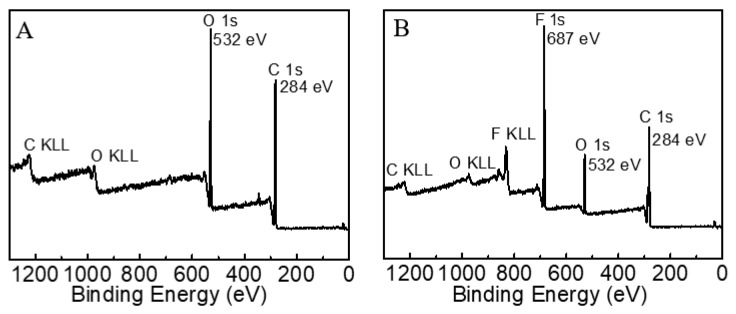
XPS full spectrum of pristine cotton fabric (**A**) and treated fabric (**B**).

**Figure 11 molecules-28-03369-f011:**
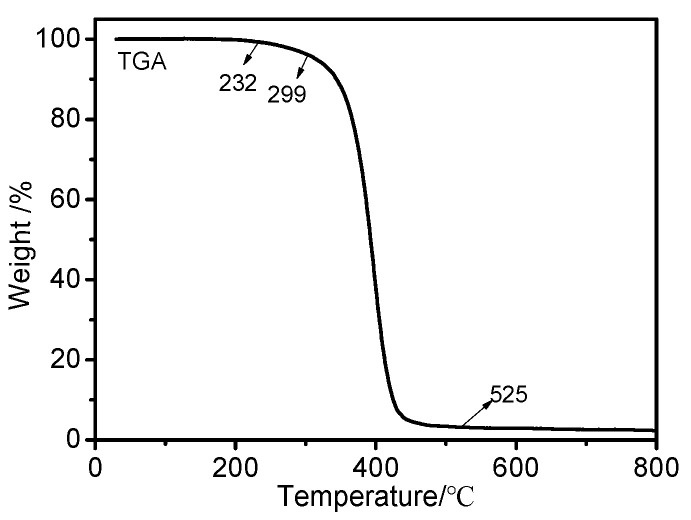
TG (10 °C/min) curve of OFAE-SA-BA under nitrogen.

**Figure 12 molecules-28-03369-f012:**
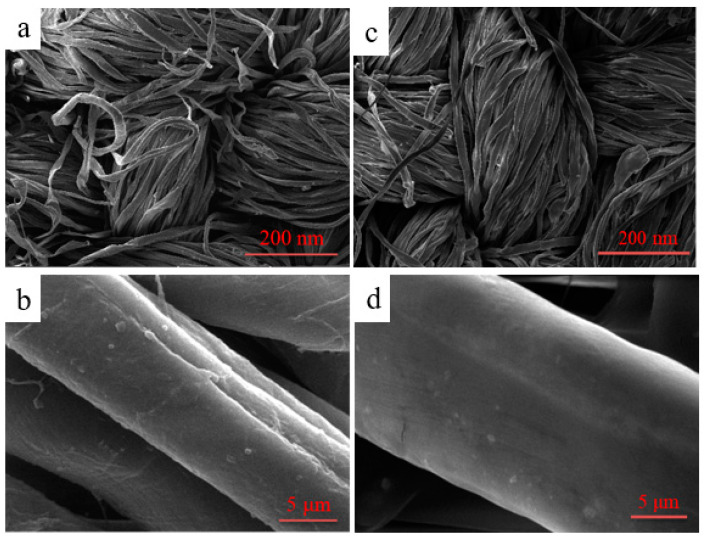
SEM images of unfinished (**a**,**b**) and finished cotton fibers (**c**,**d**).

**Figure 13 molecules-28-03369-f013:**
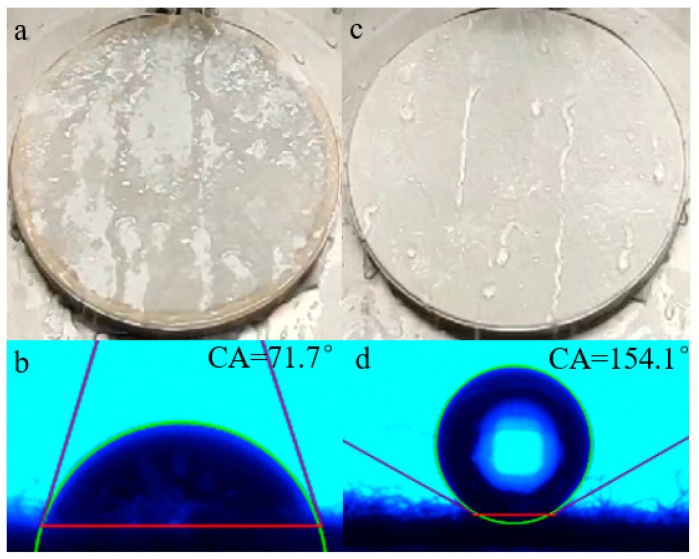
Spray property and static contact angles of untreated (**a**,**b**) and treated cotton fabric surface (**c**,**d**).

**Figure 14 molecules-28-03369-f014:**
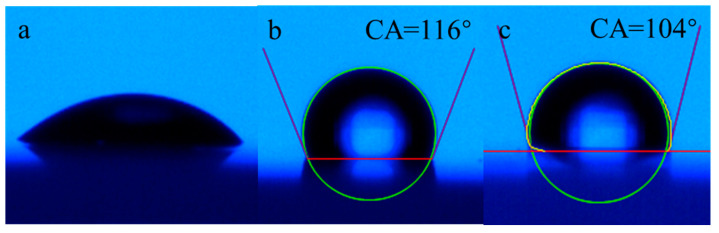
Oleophobicity of untreated and treated cotton fabric. (**a**): untreated cotton fabric finished by white mineral oil, (**b**,**c**): treated cotton finished by n-hexadecane and n-tetradecane, respectively.

**Table 1 molecules-28-03369-t001:** The content of surface functional groups ascertained by XPS.

Samples	Atomic (%)		
	C	O	F
Pristine fabric	73.47	25.67	—
Treated fabric	66.27	13.48	20.26

**Table 2 molecules-28-03369-t002:** GPC testing of OFAE-SA-BA.

Sample	Mn	Mw	PDI (Mw/Mn)
OFAE-SA-BA	345,246	363,618	1.05

**Table 3 molecules-28-03369-t003:** Durability and water repellency of cotton fabric.

Washing Times	Contact Angle/°	Moisture-Resistance Rating	Oil-Repellency Rating
0	156.7 ± 2.57	4	4
5	148.9 ± 2.02	4	4
10	145.2 ± 0.98	4	4
15	143.1 ± 1.87	4	3
20	140.4 ± 1.23	3	3
25	139.9 ± 0.88	3	3
30	139.8 ± 1.02	3	3

## Data Availability

Not applicable.

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
