# Peer review of "Synthesis and Performance Analysis of Green Water and Oil-Repellent Finishing Agent with Di-Short Fluorocarbon Chain"

_molecules, 2023, doi:10.3390/molecules28083369_

Round 1
Reviewer 1 Report
Dear authors after a carefully lecture of the article I detect some issues.
Please check line 10.
Lines 30-40. The text is redundant about the properties of fluoropolymers.
Why the authors don’t use 13C-NMR to analyze the synthetic compounds?
Lines 137-142 have format issues.
Author Response
Grammatical errors in line 10 has been revised. The text in lines 30-40 is considered redundant has been detected. Format issues in lines 137-142 has been revised.13C-NMR analysis have been added in the manuscript.
Reviewer 2 Report
The manuscript is well written and documented and the citations are suitable. The molcule has been studied for water repellency. The conclusions are supported well by results and discussion. Therefore I suggest acceptance of this manuscript after minor revision on the following.
In the introduction, the various applications of hydrophobic compounds are not explored fully. I suggest the authors to add interesting works like, Chem. Commun. 2015, 51, 14251– 14254.
Serious typo: for example, in abstract: first and second sentences
At some points, the manuscript looks like a very raw draft due to the writing is not up to the mark. A serious example is Page 2, Line 54.
Author Response
In the introduction, related papers about the applications of hydrophobic compounds have been added in the second paragraph of introduction. The full text has been carefully checked and errors have been corrected
Reviewer 3 Report
The main goal of the described research was examination of some fluorinated organic compounds as new, water repellant agents. The starting key-compound abbreviated as OFPA is a known compound and its synthesis was described in an earlier publication (J. General Chem. of the USSR 1991, 61, 611) and in a patent (SINOCHEM HOLDINGS CORPORATION LTD; SICHUAN UNIVERSITY OF SCIENCE AND ENGINEERING - CN113292403, 2021, A). None of these publications is cited by Authors.
Due to limited novelty in terms of organic synthesis, mechanisms and spectroscopic properties of the described compounds, this manuscript does not fit in the profile of Molecules. The described study should be resubmitted to another MDPI journal, e.g. Textiles or Water.
Author Response
J. General Chem. of the USSR 1991, 61, 611, we really didn't investigate this paper,this is a mistake in our work.
CN113292403, 2021, A, the patent is ours,we did not cited this petent to avert suspicion

Reviewer 4 Report
The paper is written very well, it is clear and concise. The authors explain very well in this paper about the designed and synthesized a novel fluorine containing water-repellent agents (OFAE-SA-BA).
I appreciate authors provided 1H NMR, FT-IR and XPS analysis data diagram in the manuscript. I have few comments included for the betterment of the paper.
1. I recommend to the authors, include 1H NMR data values and FT-IR data values in materials and methods section.
2. Some grammatical mistakes in the manuscript, please recheck and make corrections.
Author Response
1H NMR data values and FT-IR data values were added in materials and methods section.
The grammatical mistakes in the manuscript have been rechecked and revised.
Round 2
Reviewer 1 Report
Dear authors all my concerns were taking into account. I think that the results are improved.
Reviewer 3 Report
The revised version, supplemented by spectroscopic data slightly enhanced scientific quality/novelty of this work. However, in the case of starting monomer OFAE the 19F NMR should also be registered and discussed. In addition, description of the 13C NMR data, requires one decimal given in the chemical shift value, i.e. . 105.0-120.0 (and not 105-120), 70.5 (and not 70.52), 69.2 (and not 69.15), 165.0, 132.0 , 127.0 (page 10 for OFAE)), etc.
The recommended reference to first report on monomeric, fluorinated compounds is still missing.
My opinion presented in the first version of this review has not changed and I still think that this work does not fit in the profile of Molecules.